# Effects of Temperature on the Thermal Biology and Locomotor Performance of Two Sympatric Extreme Desert Lizards

**DOI:** 10.3390/ani15040572

**Published:** 2025-02-17

**Authors:** Yuhan Zheng, Ruichen Wu, Ziyi Wang, Xunheng Wu, Huawei Feng, Yi Yang

**Affiliations:** Xinjiang Key Laboratory for Ecological Adaptation and Evolution of Extreme Environment Biology, College of Life Sciences, Xinjiang Agricultural University, Urumqi 830052, China; zyhzyh_yuhanzheng@163.com (Y.Z.); wuwuwurc@163.com (R.W.); w291235785@163.com (Z.W.); 19861765873@163.com (X.W.); 17554223065@163.com (H.F.)

**Keywords:** climate warming, *Eremias roborowskii*, *Phrynocephalus axillaris*, thermal biology, locomotor performance, convergent evolution

## Abstract

Climate warming exerts profound influences on organisms worldwide, including their growth, development, reproduction, and genetics, ultimately leading to alterations in habitat structure, disruptions in the population cycles of organisms, and reductions in genetic diversity among species. Due to their thermosensitivity and ecological significance, lizards are ideal models for investigating animal adaptations to climate change. This study investigates the effects of temperature on the thermal biology and locomotor performance of two sympatric extreme desert lizards, *Eremias roborowskii* and *Phrynocephalus axillaris*. Our findings reveal that body size and ecological strategies drive diverse and complex thermoregulatory mechanisms in the two lizard species. Moreover, extreme temperatures impose significant constraints on locomotor performance and physiological functionality in both species. This study provides critical insights into convergent evolution and ecological adaptations in sympatric desert lizards. It also highlights key implications for biodiversity conservation and responses to global climate change.

## 1. Introduction

Climate warming, a prominent focus of global scientific research, profoundly impacts the growth, development, reproduction, and genetic diversity of organisms worldwide [1,2,3]. The ensuing high temperatures have led to drastic changes in climate patterns, such as extreme events and increased rainfall variability [4], which have led to significant alterations in habitat structure, disruptions in biological population cycles [5], and loss of the genetic diversity of species [2]. Furthermore, warming negatively impacts survival strategies, reproductive rates, and immune functions, thereby increasing the vulnerability of organisms under ongoing climate change.

As a key regulatory parameter of life activities, the stability of body temperature is crucial for animals to maintain normal physiological functions and express physiological potential at high levels [6,7]. An appropriate body temperature can ensure efficient life activities of animals, while an extreme body temperature may cause a series of physiological disorders and even endanger life. Effectively regulating and maintaining body temperature within a tolerable range under extreme weather conditions is a critical survival challenge for animals. Therefore, in the complex and ever-changing nature, maintaining the body temperature within the normal range is vital for sustaining life activities.

Compared with endotherms, reptiles’ physiological state [8], functional performance [9], reproduction level [5], distribution range, and population size [10] are closely linked to ambient temperature. Among reptiles, lizards exhibit a particularly high sensitivity and dependence on ambient temperature, as their body temperature, behavior, and basic physiological functions are directly and significantly affected by environmental conditions [11]. Nevertheless, lizards are not completely constrained by their thermal environment [12,13]. Their thermoregulatory mechanisms that consist of both behavioral and physiological responses enable them to adjust their behavioral and/or physiological limits to compensate for thermal fluctuations in the environment [14,15]. Furthermore, the Bogert effect demonstrates that animals can modify their behaviors, such as seeking shade for heat avoidance or aggregating for warmth, to reduce direct exposure to extreme environmental conditions. This behavioral adaptability enables them to respond to variations in ambient temperature and mitigate the pressures of natural selection, which reduces the reliance on physiological regulation and potentially constrains physiological evolution [16,17].

Ectotherms have a weaker physiological thermoregulatory capacity than endotherms, relying more on behavioral regulation to buffer the impact of ambient temperature fluctuations [18,19]. Some reptile species are thermal compliant animals, whose body temperature fluctuates with ambient temperature changes, giving them environmental flexibility. In contrast, the precise thermoregulators maintain body temperatures within a narrower range through active behavioral regulation [20,21]. Additionally, differences in life habits influence thermoregulatory strategies in lizards, with diurnal lizards more inclined to actively regulate their body temperature through basking, while nocturnal lizards show more thermal compliance, and the efficiency of their thermoregulation is strongly correlated with the ambient temperature [20,22].

*Eremias roborowskii* (genus *Eremias*, family Lacertidae) and *Phrynocephalus axillaris* (genus *Phrynocephalus*, family Agamidae) are two species of lizards that are distributed in the desert area of Turpan Basin in the Xinjiang Uyghur Autonomous Region of China. *Eremias roborowskii* is one of the most geographically and ecologically widespread lizard species in the genus *Eremias*. Currently, studies on *E. roborowskii* include sex chromosomes, scalation variation and subspecies classification [23], locomotor phenotypes, and morphology. Research on *P. axillaris* includes habitat suitability evaluation and corridor modeling construction [24], age identification and growth patterns [25], sexual dimorphism in morphological traits, nest site characteristics [26], and genetic structure.

Ecological niche segregation is a key mechanism for species coexistence and biodiversity. Sympatric species often differentiate through behavioral, morphological, or ecological traits, reducing competition by minimizing niche overlap [27]. With global climate change intensifying, understanding how sympatric species cope with temperature extremes has become increasingly critical [28]. Lizards are ideal models for studying the mechanisms of animal adaptation to global climate change because of their high sensitivity to temperature changes and their special position in physiological ecology [29,30,31]. In this study, the effects of climate warming on the thermal biology and locomotor performance of two sympatric extreme desert lizards were examined. Meanwhile, comparing the similarities and differences in the behavioral thermoregulation of two sympatric lizards will provide a vital supplement for the study of reptile ecological adaptability in extreme desert environments under climate warming.

## 2. Materials and Methods

### 2.1. Test Animals

A total of 7 male adult *P. axillaris* and 7 male adult *E. roborowskii* were collected in August 2024 from the Turpan region, situated in the Turpan Basin, Xinjiang Uyghur Autonomous Region, China (89°11′ E, 42°54′ N) (Figure 1). Sand from the lizards’ habitat was also collected for locomotor activity experiments. All lizards were housed in 30 cm × 21 cm × 15.5 cm (L × W × H) plastic enclosures that simulated natural habitats with soil and sand substrates, and provided with sufficient food including mealworms and calcium and vitamin-enriched water. All experimental procedures were approved by the Animal Welfare and Ethics Committee of Xinjiang Agricultural University, Urumqi, Xinjiang, China (approval number: 2023031). After the experiments, all lizards were safely returned to their natural habitat.

### 2.2. Measurement of Morphological Indexes of Lizards

Lizards were weighed using an electronic balance with a precision of 0.01 g to record mass. Snout–vent length (SVL), head length (HL), head width (HW), head depth (HD), mouth breadth (MB), axilla–groin length (AG), and abdominal length (AB) were measured using digital vernier calipers (precision: ±0.01 mm). MB (mouth breadth) was defined as the distance between the left and right corners of the mouth, while AG (axilla–groin length) was measured as the distance from the posterior margin of the forelimb to the anterior margin of the hindlimb. Other measurements included AW (abdominal width: the distance between the widest part of the abdomen and the base of the tail), FLL (forelimb length: from the forelimb base to the longest digit), HLL (hind limb length: from the hindlimb base to the longest toe), TL (tail length: distance from the cloaca to the caudal tip), FTL (fore toe length: distance from the toe base to the tip), and HTL (hind toe length: distance from the toe base to the tip). A total of 21 morphological indices were recorded for each individual [32,33].

### 2.3. Measurement of Lizard Temperature

*Eremias roborowskii* and *P. axillaris* were placed in a constant-temperature incubator (Shanghai-Heng Scientific Instrument Co., Ltd., Shanghai, China), with the environmental temperature (T_e_) set to range from 18 °C to 42 °C. Lizards were acclimated for over two hours at each temperature gradient (2 °C intervals). A thermocouple temperature recorder (Source-Hengtong YHT309S, range: −200 °C to 1370 °C; resolution: 0.01 °C) was positioned on the dorsal surface or other smooth areas of the lizards to measure their selected body temperature (T_sel_) once readings stabilized. T_sel_ are those selected by an animal in a laboratory thermal gradient that lacks thermoregulatory costs (i.e., fundamental thermal niche).

Thermal tolerance tests were conducted to measure the critical thermal maximum (CT_max_). Seven lizards from each species were sampled and placed individually in transparent plastic boxes (15 cm × 10 cm × 5 cm) within a thermostatic incubator. Thermometer conduits equipped with thermocouples were attached to the anterior cloacal region and secured with Micropore surgical tape to record body temperature at one-second intervals. Observations continued until the lizard could no longer self-right when supine, at which point CT_max_ was recorded [34,35,36]. Following CT_max_ measurement, lizards were promptly removed from the heat source and placed in a cool, ventilated area for rapid cooling to prevent overheating. After a three-day recovery period, the same seven lizards from each species were used to measure the critical thermal minimum (CT_min_). Each individual was placed in a clear plastic box and cooled in a −20 °C refrigerator, with body temperatures recorded at one-second intervals. The temperature at which the lizard was unable to self-right when supine was recorded as CT_min_ [34,35,36]. The rates of temperature change for CT_max_ and CT_min_ of *E. roborowskii* were measured at approximately 0.84 °C/min and 1.20 °C/min, respectively. Similarly, the rates of temperature change for CT_max_ and CT_min_ of *P. axillaris* were measured at approximately 0.97 °C/min and 1.31 °C/min, respectively. The temperature change rate of the constant-temperature incubator was 4.96 °C/min.

In August 2024, individuals of *E. roborowskii* and *P. axillaris* actively moving within the study area were captured randomly using insect nets. Immediate field measurements of active body temperature (T_b_) and environmental temperature (T_e_) were quickly conducted with thermometers to validate laboratory experiments. T_b_ represents the internal temperature of an animal measured in nature during its active period, reflecting its realized thermal niche. T_b_ is the temperature of the smooth part of the lizard’s dorsal, while T_e_ is the surface temperature of the area where lizards are active.

### 2.4. Measurement of Locomotor Activity of Lizards

Each *E. roborowskii* and *P. axillaris* was placed in a thermostatic incubator, with T_e_ set to a range of 18 °C to 42 °C and precisely regulated in 2 °C increments. Lizards were acclimated to each temperature gradient for at least two hours to ensure their body temperature was equilibrated with the chamber temperature to the corresponding test temperature. A transparent acrylic runway (100 cm × 20 cm × 30 cm, L × W × H) was prepared, with a black oil-based marker used to draw a 20 cm line on the side and bottom of the runway. Sand and gravel were placed inside the runway to simulate the lizards’ natural movement environment, enhancing the ecological validity of the experimental results. Once the lizards’ body temperature was stabilized, they were promptly transferred from the incubator and released at the track’s starting point, then guided to run along the track to the endpoint. If the lizards paused, non-invasive methods involving gently brushing the tail base were applied to encourage them to resume running, ensuring the integrity of the test. A high-definition video camera recorded the lizards’ movements, capturing details such as the number of pauses and distances covered during continuous runs, ensuring data accuracy and reproducibility.

Each lizard was tested twice, with several hours of recovery between tests. The fastest run of the two tests, covering a distance of at least 20 cm, was recorded as the sprint speed (V_max_, m/s). A complete locomotor test was defined as the lizard fully traversing the runway and turning around. The best performance from the two test trials was used for subsequent experimental analysis.

Pauses occurring at the turnaround point were excluded from the analysis. The maximum movement distance was defined as the longest uninterrupted run achieved within a single test.

### 2.5. Data Analysis

All experimental data were statistically and graphically analyzed using IBM SPSS Statistics 21, GraphPad Prism 10.2.3, and OriginPro 2024. Kolmogorov–Smirnov and Levene’s tests were employed to assess data normality and homogeneity of variance prior to statistical analyses. Experimental data were presented as mean ± SD, with statistical significance set at *p* < 0.05. Morphological differences between the two lizard species were analyzed using independent samples *t*-tests, non-parametric tests, and multivariate analysis of covariance (MANCOVA) with Bonferroni correction.

Principal component analysis (PCA) was applied to further explore variation in morphological traits. A one-way linear regression model was fitted to analyze the relationship between T_e_ and T_sel_. Differences in T_e_, sprint speed, number of pauses, and maximum movement distance were analyzed using one-way ANOVA and Duncan’s Multiple Range Test (DMRT).

## 3. Results

### 3.1. Differences in Morphological Characteristics of Two Sympatric Lizards

As shown in Table 1, a significant difference in body size was observed between *E. roborowskii* and *P. axillaris* individuals (*t* = 10.719, *df* = 12, *p* < 0.05), with *E. roborowskii* exhibiting a significantly greater head–body length (65.53 ± 2.86 mm, n = 7) compared to *P. axillaris* (44.83 ± 4.53 mm, n = 7).

MANCOVA with Bonferroni correction indicated that *E. roborowskii* was significantly larger than *P. axillaris* in SVL, MASS, AG, TL, TBW, D1, D2, T1, T3, T4, and T5 (SVL: *p* < 0.05; MASS: *p* < 0.05; AG: *p* < 0.05; TL: *p* < 0.05; TBW: *p* < 0.05; D1: *p* < 0.05; D2: *p* < 0.05; T1: *p* < 0.05; T3: *p* < 0.05; T4: *p* < 0.05; T5: *p* < 0.05). Meanwhile, *E. roborowskii* did not differ significantly from *P. axillaris* in HL (*p* > 0.05), HW (*p* > 0.05), HD (*p* > 0.05), MB (*p* > 0.05), AW (*p* > 0.05), FLL (*p* > 0.05), HLL (*p* > 0.05), D3 (*p* > 0.05), D4 (*p* > 0.05), D5 (*p* > 0.05), and T2 (*p* > 0.05) (Table 1). Meanwhile the first three principal components (PCs) derived from the PCA each contributed more than 10%, cumulatively explaining 78.9% of the morphological differences between the two sexes (Table A1).

Morphological variables such as TL, AG, SVL, MASS, and TBW were positively loaded in the first principal component (PC1), while MB and AW were negatively loaded. The second principal component (PC2) was positively loaded with HW, AW, MB, FLL, and HD, but negatively loaded with TL, SVL, MASS, and AG (Figure 2). The third principal component (PC3) was positively loaded with AW, HW, MASS, TBW, and AG, but negatively loaded with FLL, HLL, HL, and SVL (Table A1). Therefore, *E. roborowskii* exhibits greater appendage length, body size, body weight, and reproduction-related traits (e.g., axillary span and abdominal width) compared to *P. axillaris*.

### 3.2. Correlation Analysis of T_e_ and T_sel_ of Two Sympatric Lizards

The T_e_ range (18–42 °C, with a 2 °C temperature gradient) was used to measure the T_sel_ of *E. roborowskii* during a stepwise warming process. Results showed a significant positive correlation between T_sel_ (Y) and T_e_ (X) for *E. roborowskii*: Y = 0.8344X + 4.532 (F = 506.2, *p* < 0.0001, R^2^ = 0.9787). The slope of the T_sel_ line (*k* = 0.8344) indicated that for every 1.0 °C increase in T_e_, T_sel_ increased by 0.8344 °C (Figure 3A). The T_sel_ line intersected the isotherm at 27.37 °C, indicating equilibrium between T_sel_ and T_e_ for *E. roborowskii* at this temperature. The absolute difference between T_e_ and T_sel_ for *E. roborowskii* increased gradually both above and below the equilibrium temperature (27.37 °C).

Similarly, T_e_ ranged from 18 to 42 °C (temperature gradient: 2 °C) during gradual warming, and the T_sel_ of *P. axillaris* was measured. The relationship between T_sel_ (Y) and T_e_ (X) of *P. axillaris* showed a significant positive correlation: Y = 0.7582X + 6.539 (F = 464.8, *p* < 0.0001, R^2^ = 0.9769). The slope of the T_sel_ line (*k* = 0.7582) indicated that for every 1.0 °C increase in T_e_, T_sel_ increased by 0.7582 °C (Figure 3B). The T_sel_ line intersected the isotherm at 27.04 °C, indicating equilibrium between T_sel_ and T_e_ for *P. axillaris* at this temperature. The absolute difference between T_sel_ and T_e_ for *P. axillaris* increased gradually both above and below the equilibrium temperature (27.04 °C).

### 3.3. Effect of Temperature on Locomotor Performance in Two Sympatric Lizards

Sprint speed, number of pauses, and maximum movement distance for *E. roborowskii* and *P. axillaris* were measured across a T_e_ range of 18 to 42 °C (temperature gradient: 2 °C). The results indicated that the sprint speed of *E. roborowskii* increased with T_e_ from 18 to 30 °C, peaking at 28 to 30 °C. However, as T_e_ rose beyond 30 °C, sprint speed decreased gradually until 42 °C. Similarly, *P. axillaris* exhibited increasing sprint speed from 18 to 32 °C, peaking at a slightly higher T_e_ range of 30 to 32 °C. Furthermore, an inverse correlation was observed between T_e_ and sprint speed when the T_e_ increased from 32 to 42 °C (Figure 4A).

For the number of pauses, a gradual decrease was observed in *E. roborowskii* as the T_e_ increased from 18 °C to 30 °C, reaching a minimum at 30 °C. Beyond 30 °C, the number of pauses gradually increased. However, the impact of T_e_ on the number of pauses in *P. axillaris* was less than that of *E. roborowskii*, with a minimum observed at 28 °C (Figure 4B).

For maximum movement distance, *E. roborowskii* exhibited a clear increasing trend from 18 to 32 °C. Beyond 32 °C, maximum movement distance began to decline. Similarly, *P. axillaris* showed an increasing trend in maximum movement distance from 18 to 34 °C. Beyond 34 °C, maximum movement distance for *P. axillaris* gradually decreased (Figure 4C).

## 4. Discussion

This study conducted MANCOVA and PCA to analyze 21 indicators, including the morphological characteristics of *E. roborowskii* and *P. axillaris*, with the aim of identifying factors that may influence their locomotor performance. Subsequently, the study measured the T_sel_, T_b_, CT_min_, CT_max_, and locomotor performance of the two sympatric lizards under Te, followed by a comprehensive analysis of their thermal performance. The results demonstrated that temperature significantly affects the locomotor fitness and physiological functions of two sympatric lizards. Furthermore, body size and ecological strategies were found to drive the diverse and complex thermal regulation mechanisms.

*Eremias roborowskii* and *P. axillaris* are lizards endemic to China, inhabiting the desert area of Turpan Basin [30]. Both species are diurnal, exhibiting daily activity rhythms compatible with high daytime temperatures, particularly engaging in basking and foraging activities in the pre-noon hours. In the late afternoon, they seek shelter and rest due to high temperatures, while at night, they conserve energy and avoid predators by remaining hidden [37,38]. Despite minor differences in thermoregulation, these two sympatric desert lizards exhibit significant variations in body size and locomotor abilities, which might be due to the different predatory strategies. *Eremias roborowskii*, an ambush predator, typically inhabits concealed sites like rock crevices and grasses, allowing it to hide swiftly when feeding or evading predators. When prey is detected, it relies on its concealment to launch sudden, rapid attacks to capture prey [39,40]. Conversely, *P. axillaris* is an active predator that aggressively chases and attacks prey during feeding [39]. These distinct ecological and behavioral strategies contribute to their ability to coexist within the same environment.

Through natural selection, biological evolution optimizes the balance between morphology and ecology, with locomotion playing a crucial role in ecologically relevant activities such as mating, foraging, territorial defense, and predator avoidance [41]. Locomotion is a key factor, indicator, and measure of animal adaptation [42]. In addition, it has been shown that locomotor performance in lizards is strongly influenced by body size, body weight, and reproductive status [43,44]. Among locomotor performances, sprint speed is commonly used as an indicator of fitness and the ability to survive predation pressure [45], while endurance serves as an indicator of foraging strategies, mate searching, and predation evasion [46,47]. Morphology also significantly impacts locomotor performance [48]. For example, appendages play a decisive role in determining locomotor traits, such as maximum sprint speed, in lizards [49]. SVL, HLL, and HTL are strongly correlated with lizard running speed [50]. The tail, which constitutes a significant portion of a lizard’s body, functions as a balance organ for locomotion and a major site for energy storage [51]. MANCOVA revealed that the traits such as TL, SVL, MASS, TBW, and AG of *E. roborowskii* were significantly higher than those of *P. axillaris*, suggesting that *E. roborowskii* possessed a morphological advantage in sprinting ability.

Global climate change has become one of the most serious threats to biodiversity, especially for metazoans, whose physiological functions are closely linked to environmental conditions such as temperature [14]. It has been shown that locomotor performance in particular is highly sensitive to body temperature [52]. Lizards are able to maintain their body temperature within a certain range through behavioral thermoregulation, which is one of their effective mechanisms for mitigating temperature extremes [53]. The intersection of the body temperature fitting line and the isotherm in metameric animals represents the equilibrium point of heat exchange between body temperature and the environment [54]. Our study found that the equilibrium point of heat exchange was 27.37 °C for *E. roborowskii*, slightly higher than 27.04 °C for *P. axillaris*. The correlation coefficient (linear slope *k*) reflects the relationship between body temperature and T_e_. Stronger thermoregulatory ability is indicated by a linear slope closer to 0, while weaker regulation is reflected by a slope closer to 1 [55]. The absolute difference between the correlation coefficient of *E. roborowskii*’s T_sel_ line (*k* = 0.8344) and the isotherm (*k* = 1) was 0.1656, smaller than that of *P. axillaris* (*k* = 0.7582), which was 0.2418. Therefore, in the absence of shelter or any external heat source, lizards placed in a thermostatic incubator were restricted in their activities, relying solely on physiological thermoregulation. This observation indicates that the correlation coefficient between T_sel_ and T_e_ is relatively smaller in *P. axillaris*, suggesting a lesser dependence on T_e_ and a greater degree of physiological regulation.

Temperature exerts a notable influence on the locomotor performance of lizards. Within the thermal tolerance range, as the ambient temperature rises, the lizard’s body temperature approaches the optimal level, leading to a gradual enhancement in physiological performance. However, once the optimal temperature is surpassed, the behavioral performance of lizards declines rapidly [55]. Furthermore, when the lizards’ morphological structure constrains the maximization of all types of locomotor performance, a possible trade-off among these locomotor capabilities emerges [56]. The two sympatric lizards achieved peak sprint speeds within a similar T_e_ range, with no significant differences observed at 20–24 °C, 30 °C, and 34–38 °C. However, *E. roborowskii* showed significantly higher sprint speeds than *P. axillaris* at 26–28 °C. Notably, *P. axillaris* was faster at extreme temperature gradients of 18 °C, 40 °C, and 42 °C, indicating that *P. axillaris* exhibits superior physiological regulation under extreme temperature conditions, such as 42 °C. Specifically, lizards with higher sprint speeds tend to tire quickly and achieve only moderate endurance, whereas individuals with greater endurance generally exhibit lower sprint speeds, enabling longer runs [57]. The trade-off between sprint speed and endurance has been validated in several species of squamate reptiles, such as *Lacerta vivipara*, which is a result of long-term evolution through morphological and physiological considerations, and is closely related to factors such as growth, survival, and predation [58,59]. However, this trade-off is not universal, as studies on 14 lizard species from North America did not identify this relationship [57].

Although differences between the two species were not significant, *E. roborowskii* exhibited fewer pauses than *P. axillaris* in the T_e_ range of 26–34 °C, which is close to the optimal range for locomotor performance, with significant differences observed at 30 °C and 34 °C. Endurance sets an upper limit on activities requiring speed and consistency, potentially restricting a species’ survival within its habitat range [60]. This suggests that *E. roborowskii* demonstrates superior endurance traits. For maximum movement distance, *P. axillaris* possessed a significantly shorter distance than *E. roborowskii* at 18–20 °C, whereas the reverse was observed at 26–34 °C. *Eremias roborowskii* demonstrated optimal locomotor performance at approximately 30–32 °C, whereas *P. axillaris* exhibited peak performance at slightly higher T_e_ of 32–34 °C. These findings suggest that the two lizards exhibit distinct temperature adaptations, which may be associated with differences in their ecological niches and physiological traits. *Eremias roborowskii* achieved peak sprint speeds, with fewer pauses, and exhibited longer continuous movement distance at slightly lower T_e_, which can be attributed to its advantageous body size and ambush-type predation strategy. In contrast, *P. axillaris* performs better in sprint speed and maximum movement distance, with slightly weaker endurance at higher T_e_, due to a trade-off between sprint speed and endurance and the need for a more proactive foraging strategy.

The disparity in the speed–endurance trade-off between the two sympatric lizard species is likely driven by differences in their limb morphology and SVL. Longer limbs are generally associated with greater stride length, which reduces the number of steps needed to cover a given distance, thereby enhancing energy efficiency during sustained locomotion [61]. The ability of *E. roborowskii* to sustain prolonged activity with minimal pauses may be due to the biomechanical advantages of its longer limbs that optimize energy efficiency and enhance heat dissipation during locomotion. Additionally, longer limbs may improve leverage and stability, further supporting endurance in challenging environments [62]. In contrast, the ability of *P. axillaris* to achieve higher sprint speeds and longer burst distances in higher temperatures, especially more in extreme temperature conditions, may be due to its shorter limbs. Shorter limbs are typically associated with increased stride frequency due to their reduced moment of inertia, enabling rapid acceleration and faster limb cycling, which is particularly advantageous for sprinting where quick bursts of speed are essential [44,63]. However, the trade-off is reduced endurance, as shorter limbs tend to require higher energy expenditure per unit distance, making sustained locomotion less efficient [64]. Similar to other lizards, a distinct relationship exists between limb length and locomotor performance. Arboreal lizards, characterized by long, slender limbs, are good at long-distance branch movement and quick direction changing for prey or predator avoidance [62]. Ground-dwelling lizards with short, stout limbs are better adapted for rapid dashes across open areas and can quickly hide when threatened [63]. In conclusion, limb morphology may have evolved to optimize energy utilization strategies, tailored to the specific demands of different ecological niches.

Global warming and rising ambient temperatures may significantly impact the locomotor performance of these two sympatric lizards. In the Turpan region, extreme surface temperatures during July and August have been recorded as exceeding 70 °C, occasionally reaching as high as 83 °C [65]. Moreover, the T_b_ of these lizards has exceeded the optimal range for locomotor performance, approaching their CT_max_ (Table A2 and Table A3), indicating exposure to extreme thermal conditions. T_b_ is a critical factor with strict upper and lower limits for survival and physiological function in metazoans. Although ectotherms experience a broad range of T_e_, T_b_ must remain within physiological limits for survival at all developmental stages [66]. *Phymaturus zapalensis* and *P. querque* in high-temperature environments avoided overheating by selecting temperatures below T_sel_ [67]. This suggests that lizards use behavioral strategies and habitat microclimates to mitigate the effects of extreme climatic conditions, such as seeking shade during hot summers and burrows during cold winters [19]. However, the energetic costs of thermoregulation may compromise vital functions such as reproduction, foraging, and social displays [68,69], and combined with risks of overheating and predation, may limit precise thermoregulation. This indicates that their locomotor performance in natural habitats is often suboptimal. Nonetheless, *P. axillaris* individuals exhibited significantly higher T_b_ during field activities than *E. roborowskii*, suggesting that *P. axillaris* may achieve optimal speeds more effectively in the wild.

## 5. Conclusions

In summary, *E. roborowskii*’s T_sel_ line intersected isotherm is higher than that of *P. axillaris*, and the difference in correlation coefficients between the T_sel_ line and isotherm indicates that *P. axillaris* possess a superior physiological thermoregulatory capacity and less dependence on T_e_. Meanwhile, *P. axillaris* and *E. roborowskii* showed distinct strengths in sprint speed, number of pauses, and maximum distance movement. *P. axillaris* demonstrated better sprint speed and continuous movement distance at higher T_e_, while *E. roborowskii* showed better endurance (fewer pauses) and continuous movement distance, which might be related to their body size, predation strategy, and thermotolerance (Figure 5). However, if ambient temperatures continue to rise at the current rate, both species may face survival challenges in the future. Despite their physiological/behavioral thermoregulatory functions, extreme temperatures may still be detrimental to their locomotor performance and other critical physiological functions. Therefore, to accurately assess the impact of climate warming on these two species of sympatric lizards, it is essential to monitor physiological changes such as resting metabolic rate, evaporative water loss, energy assimilation, and liver oxidative stress at various T_e_ to determine the thermal physiological plasticity, which is crucial for exploring their adaptive capacity to global warming.

## Figures and Tables

**Figure 1 animals-15-00572-f001:**
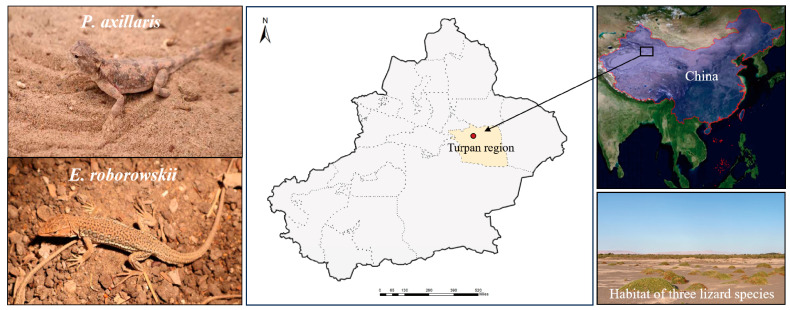
Two sympatric lizards and their habitats.

**Figure 2 animals-15-00572-f002:**
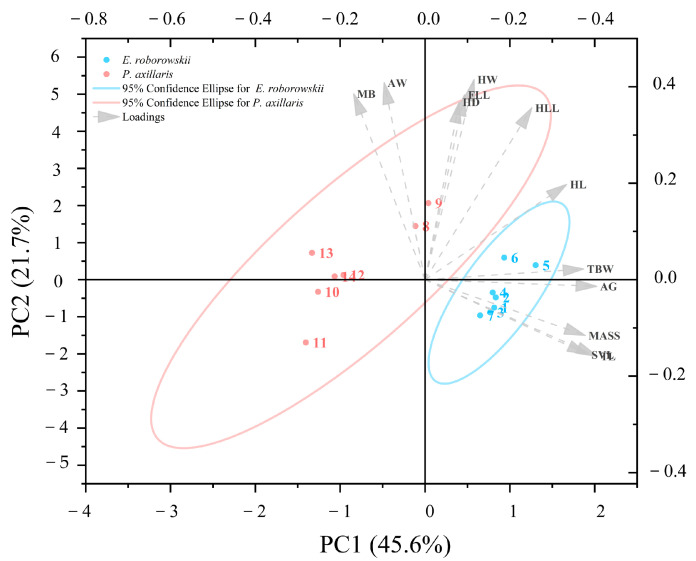
PCA plot of morphological characters of *E. roborowskii* and *P. axillaris*.

**Figure 3 animals-15-00572-f003:**
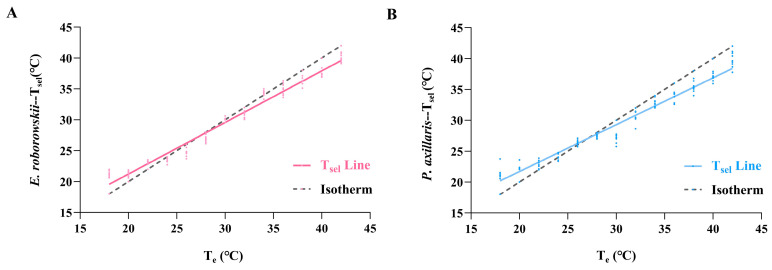
Linear regression relationships of T_e_ and T_sel_ in *E. roborowskii* (**A**) and *P. axillaris* (**B**).

**Figure 4 animals-15-00572-f004:**
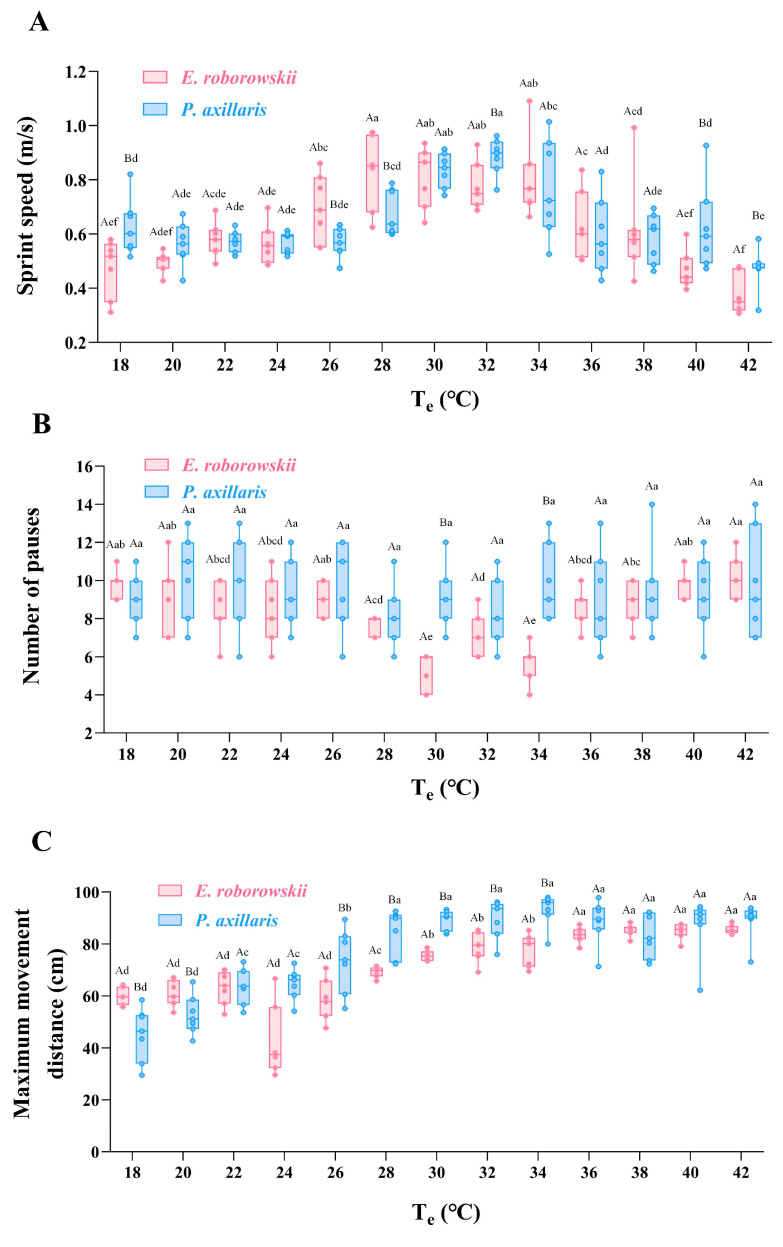
Sprint speed (**A**), number of pauses (**B**), and maximum movement distance (**C**) of two sympatric lizards at varying T_e_. Note: Upper case letters represent multiple comparison results between groups; lower case letters represent multiple comparison results within groups.

**Figure 5 animals-15-00572-f005:**
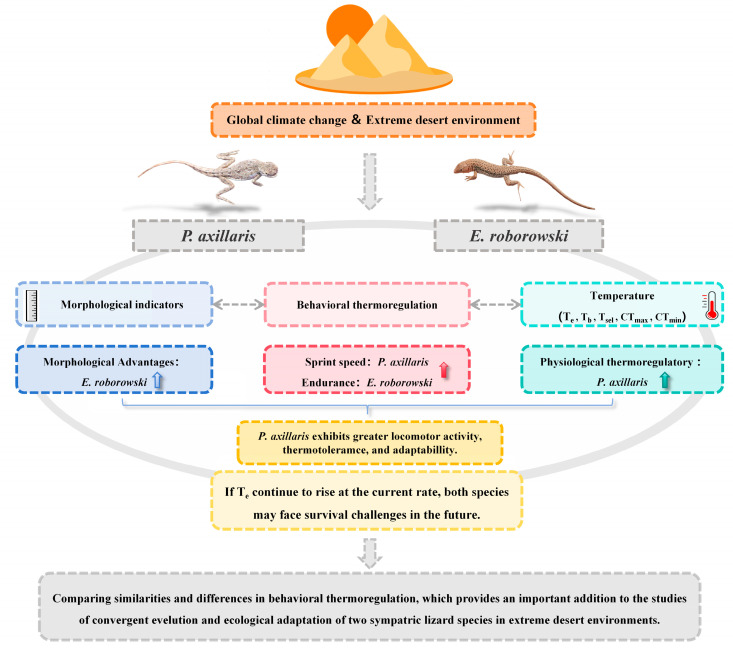
Model diagram illustrating the thermal biology and locomotor performance mechanisms of two lizard species for adaptation to the extreme desert environment.

**Table 1 animals-15-00572-t001:** Descriptive statistics on morphological traits of *E. roborowskii* and *P. axillaris*.

Traits (mm/g)	*E. roborowskii*	*P. axillaris*	*t*-Level	*p*-Value
SVL	65.53 ± 2.86 ^a^	44.83 ± 4.53 ^b^	10.719	<0.00
MASS	8.61 ± 1.18 ^a^	4.42 ± 0.90 ^b^	7.445	<0.00
HL	16.59 ± 1.48 ^a^	13.92 ± 2.99 ^a^	2.117	0.056
HW	11.90 ± 1.12 ^a^	11.73 ± 1.11 ^a^	0.281	0.784
HD	8.60 ± 1.61 ^a^	8.70 ± 1.22 ^a^	−0.125	0.902
MB	9.39 ± 0.72 ^a^	10.08 ± 0.65 ^a^	−1.883	0.084
AW	13.42 ± 0.70 ^a^	14.93 ± 2.58 ^a^	−1.499	0.160
AG	33.06 ± 2.76 ^a^	23.38 ± 2.46 ^b^	6.931	<0.00
TL	128.56 ± 2.96 ^a^	68.59 ± 8.17 ^b^	18.450	<0.00
TBW	8.31 ± 0.30 ^a^	6.41 ± 1.09 ^b^	4.458	0.001
FLL	22.75 ± 1.76 ^a^	23.45 ± 4.11 ^a^	−0.415	0.685
HLL	37.32 ± 1.68 ^a^	35.06 ± 6.39 ^a^	0.905	0.383
D1	3.59 ± 0.51 ^a^	2.21 ± 0.42 ^b^	5.603	<0.00
D2	5.38 ± 0.47 ^a^	3.57 ± 0.54 ^b^	6.707	<0.00
D3	5.51 ± 0.74 ^a^	5.01 ± 0.38 ^a^	1.597	0.136
D4	6.40 ± 1.20 ^a^	6.53 ± 0.57 ^a^	−0.258	0.800
D5	4.48 ± 0.48 ^a^	4.00 ± 1.37 ^a^	0.867	0.403
T1	3.62 ± 0.99 ^a^	2.27 ± 0.72 ^b^	2.921	0.013
T2	4.84 ± 0.88 ^a^	3.90 ± 0.97 ^a^	1.894	0.083
T3	7.25 ± 0.65 ^a^	6.03 ± 0.82 ^b^	3.096	0.009
T4	11.00 ± 0.62 ^a^	8.99 ± 1.17 ^b^	4.024	0.002
T5	7.45 ± 0.77 ^a^	5.55 ± 0.74 ^b^	4.680	0.001

Note: Values are means ± SD; different letters indicate significant differences (*p* < 0.05).

## Data Availability

Full data are available in the References list and Appendix A.

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
