# Peer review of "Effects of Temperature on the Thermal Biology and Locomotor Performance of Two Sympatric Extreme Desert Lizards"

_animals, 2025, doi:10.3390/ani15040572_

Round 1

Reviewer 1 Report (Previous Reviewer 2)

Comments and Suggestions for Authors

The revised manuscript, "Effects of Temperature on the Thermal Biology and Locomotor Performance of Two Sympatric Extreme Desert Lizards," demonstrates significant enhancements in structure, clarity, and scientific rigor. Here you can find my evaluation of the modifications and improvements relative to the first version.

Major Improvements

a) The updated title, "Effects of Temperature on the Thermal Biology and Locomotor Performance of Two Sympatric Extreme Desert Lizards," is more specific and emphasizes the study's particular focus. The previous title was somewhat ambiguous concerning "climate warming" as the primary factor; the revised version highlights the direct impact of temperature on thermal biology and locomotion.

b) The abstract has been enhanced with additional quantitative details, including body temperatures, sprint speeds, and movement patterns, thereby improving clarity for readers.

c) The revised introduction offers improved context regarding the suitability of lizards as models for investigating temperature adaptation. The manuscript now includes additional discourse on climate-induced habitat alterations, thermoregulation, and evolutionary consequences, thereby enhancing the logical coherence.

d) The implementation of thermocouple sensors for real-time body temperature measurements represents a notable methodological advancement. The incorporation of MANCOVA for morphological traits significantly improves the statistical analysis of variations in body size and locomotion.

e) Detailed comparisons of Tsel, Tb, CTmin, and CTmax among the species are presented. The regression analysis between Tsel and Te provides important insights into the thermoregulatory strategies of each species. The presentation of paired statistical tests and MANCOVA results has been improved, enhancing the rigor of the analysis.

f) The discussion on ecological niche segregation has been enhanced. The updated version provides a clearer explanation of how ecological strategies influence the observed thermal and locomotor adaptations. Increased focus is placed on morphological constraints and the trade-offs between sprint speed and endurance.

The revised manuscript shows considerable improvement; however, several areas remain that could benefit from further refinement.

a) IN ABSTRACT: It is advisable to include one or two additional numerical results, such as the values of sprint speeds.

b) TABLE 1: Table-1 with morphological traits are dense—some columns could be removed for better readability (i.e.  F-level and T-level).

c) Some sentences remain excessively lengthy and intricate, hindering readability. Dividing them into shorter sentences would enhance clarity. Minor grammatical errors remain, such as "This study contributes significantly to understanding convergent evolution and ecological adaptations in sympatric lizards inhabiting extreme desert ecosystems, as well as an important for the conservation of biodiversity and the response to global climate change.". Suggestion: "This study provides critical insights into convergent evolution and ecological adaptations in sympatric desert lizards. It also highlights key implications for biodiversity conservation and responses to global climate change." 

d) Morphology & Locomotion Trade-offs: The discussion indicates that body size, limb length, and tail length affect locomotor performance. The functional implications of these traits require further clarification.

1. The manuscript indicates that E. roborowskii possesses longer limbs and a longer tail, conferring an advantage in endurance and reducing the frequency of pauses. P. axillaris demonstrates greater speed at extreme temperatures, yet exhibits reduced endurance. The trade-off between speed and endurance is acknowledged, yet its biomechanical implications are not thoroughly elucidated.

i) Explain why longer limbs contribute to endurance.  

ii) Discuss why shorter limbs might favor sprint speed:

2. The revised manuscript discusses the classic speed versus endurance trade-off; however, it would be enhanced by incorporating comparisons with other lizard species.

This manuscript provides significant insights into the thermal biology and locomotor performance of sympatric desert lizards, enhancing our understanding of ecological adaptations in the context of climate change. The study is well-structured and scientifically relevant; however, addressing the suggested revisions—specifically clarifying locomotor trade-offs, strengthening the discussion on morphological adaptations, and refining data presentation—will enhance its impact. I suggest minor revisions to enhance clarity and completeness prior to publication.

Author Response

We appreciate the reviewer’s and editor comments that are really helpful to improve the quality of our manuscript. We have revised the manuscript according to the reviewer’s comments. Additionally, we have revised and refined the language of the manuscript for clarity and precision. All modifications in the article are highlighted in red font.

Reviewer:

The revised manuscript shows considerable improvement; however, several areas remain that could benefit from further refinement.

Abstract

Comment1. It is advisable to include one or two additional numerical results, such as the values of sprint speeds.

Response: Thank you for your suggestion to include additional numerical results in the abstract. We agree that specific data can enhance the presentation of our findings. “Eremias roborowskii demonstrated better endurance with fewer pauses and more consistent length of continuous movement at higher Te, while P. axillaris exhibited faster sprint speed (0.8355 vs 0.8157 m/s at 30℃) and greater movement distance (78.53 vs 89.82 cm at 32℃)” (L43-46)

Comment2. TABLE 1: Table-1 with morphological traits are dense—some columns could be removed for better readability (i.e. F-level and T-level).

Response: Thanks for your suggestion. We appreciate your feedback and know the importance of clear data presentation. After careful consideration we have decided to delete the F-level from the Table1.

Comment3. Some sentences remain excessively lengthy and intricate, hindering readability. Dividing them into shorter sentences would enhance clarity. Minor grammatical errors remain, such as "This study contributes significantly to understanding convergent evolution and ecological adaptations in sympatric lizards inhabiting extreme desert ecosystems, as well as an important for the conservation of biodiversity and the response to global climate change.". Suggestion: "This study provides critical insights into convergent evolution and ecological adaptations in sympatric desert lizards. It also highlights key implications for biodiversity conservation and responses to global climate change."

Response: Thanks for the reviewer’s suggestion. We have revised the sentences. “This study provides critical insights into convergent evolution and ecological adaptations in sympatric desert lizards. It also highlights key implications for biodiversity conservation and responses to global climate change.” (L25-28). Additionally, we have revised and refined the language of the manuscript for clarity and precision.

Comment4. Morphology & Locomotion Trade-offs: The discussion indicates that body size, limb length, and tail length affect locomotor performance. The functional implications of these traits require further clarification.

  1. The manuscript indicates that roborowskiipossesses longer limbs and a longer tail, conferring an advantage in endurance and reducing the frequency of pauses. P. axillaris demonstrates greater speed at extreme temperatures, yet exhibits reduced endurance. The trade-off between speed and endurance is acknowledged, yet its biomechanical implications are not thoroughly elucidated.
  2. i) Explain why longer limbs contribute to endurance.
  3. ii)Discuss why shorter limbs might favor sprint speed.

Response: Thanks for the reviewer’s comment. We have made some revisions to this part of the article to make it clearer. “The disparity in the speed-endurance trade-off between two sympatric lizard species is likely driven by differences in their limb morphology and SVL. Longer limbs are generally associated with greater stride length, which reduces the number of steps needed to cover a given distance, thereby enhancing energy efficiency during sustained locomotion [61]. The ability of E. roborowskii to sustain prolonged activity with minimal pauses may be due to the biomechanical advantages of its longer limbs that optimize energy efficiency and enhance heat dissipation during locomotion. Additionally, longer limbs may improve leverage and stability, further supporting endurance in challenging environments [62]. In contrast, the ability of P. axillaris to achieve higher sprint speeds and longer burst distances in higher temperatures, especially more in extreme temperature conditions, may be due to its shorter limbs. Shorter limbs are typically associated with increased stride frequency due to their reduced moment of inertia, enabling rapid acceleration and faster limb cycling, which is particularly advantageous for sprinting where quick bursts of speed are essential [63-64]. However, the trade-off is reduced endurance, as shorter limbs tend to require higher energy expenditure per unit distance, making sustained locomotion less efficient [65]” (L411-427).

  1. The revised manuscript discusses the classic speed versus endurance trade-off; however, it would be enhanced by incorporating comparisons with other lizard species.

Response: Thank you for reviewer’s suggestion. We have addressed this point by incorporating comparisons with other lizard species in the revised manuscript. “Similar to other lizards, a distinct relationship exists between limb length and locomotor performance. Arboreal lizards, characterized by long, slender limbs, are good at long-distance branch movement and quick direction-changing for prey or predator avoidance [62]. Ground-dwelling lizards with short, stout limbs are better adapted for rapid dashes across open areas and can quickly hide when threatened [63]. In conclusion, limb morphology may have evolved to optimize energy utilization strategies, tailored to the specific demands of different ecological niches” (L427-434).

Comment5. This manuscript provides significant insights into the thermal biology and locomotor performance of sympatric desert lizards, enhancing our understanding of ecological adaptations in the context of climate change. The study is well-structured and scientifically relevant; however, addressing the suggested revisions—specifically clarifying locomotor trade-offs, strengthening the discussion on morphological adaptations, and refining data presentation—will enhance its impact. I suggest minor revisions to enhance clarity and completeness prior to publication.

Response: Thanks for the reviewer’s comment and constructive feedback. In the revised manuscript, we have addressed the suggested revisions by clarifying the locomotor trade-offs, strengthening the discussion on morphological adaptations, and refining the presentation of our data. For the locomotor trade-offs, we have added a detailed explanation. Regarding morphological adaptations, we have conducted a more in-depth analysis.

Reviewer 2 Report (Previous Reviewer 3)

Comments and Suggestions for Authors

This is my second review of the paper by Zheng et al. on thermal biology of two desert-dwelling lizards. This version is a distinct improvement. I especially applaud the authors for adding the ANCOVA to their analyses. However, I still fail to see a strong connection between sympatric coexistence and potential thermal responses to climate change. If the authors feel that this connection is important, they should introduce the topic of sympatric coexistence in the Introduction and not wait until the Discussion to do so.

Comments on the Quality of English Language

I have some relatively minor suggestions to improve the English. They follow line numbers given below.

17        change “Due to the” to “Due to “their”

34-38   This sentence is long and confusing with too many comma-separated clauses.

44 and other places     Never begin a sentence with an abbreviation.

47        delete comma after “size”

80-83   This is another cumbersome sentence. Try this: “Their thermoregulatory mechanisms that consist of both behavioral and physiological responses enable them to adjust their behavioral and/or physiological limits to compensate for thermal fluctuations in the environment.”

85        add “in” after “extremes”

90        Because “endotherms” is used in line 91, replace “Poikilotherms” with “Ectotherms,” which is the opposite of endotherms. Poikilotherm is the opposite of homeotherm.

92-93   change to “Some reptile species are thermal compliant animals, whose body temperature fluctuates with ambient temperature changes, giving them environmental flexibility. In contrast,…”

103      change “have been” to “are”

107      change “morphological” to “morphology”

108      change “include” to “includes”

115-118           This sentence is not clear. The introduction of a new topic, i.e., convergence evolution, clouds the issue.

127      add “and” after “substrates”

150      change “lizards” to “lizard”

182      change “represent” to “represents”

191      change “equilibrating” to “were equilibrated”

199      change to “methods involving gently brushing the tail”

205      change “lizards” to “lizard”

246      change “Meanwhlie” to “Meanwhile”

309      change “mechanisms” to “mechanism”

314      change to “change, it is becoming…”

327-328 This is a clause, not a complete sentence. Change to: “…daytime temperatures, particularly engaging in…”

366      change “reflecting” to “reflects” [Verbal forms ending in -ing are present participles and cannot be used as the primary verb to indicate action.]

380-381           change to “However, once the optimal temperature is surpassed, behavioral performances of lizards declines rapidly.          

382      change to “constraints” to “constrains”

383      change to “…performance, a possible trade-off among these…”

391      change “populations” to “individuals”

398      delete “For the number of pauses,” Begin the sentence with “Although…”

408      change “suggested” to “suggest”

409      add “be” after “may”

418      add “these” before “two”

Author Response

We appreciate the reviewer’s and editor comments that are really helpful to improve the quality of our manuscript. We have revised the manuscript according to the reviewer’s comments. Additionally, we have revised and refined the language of the manuscript for clarity and precision. All modifications in the article are highlighted in red font.

Reviewer:

Comment1. This is my second review of the paper by Zheng et al. on thermal biology of two desert-dwelling lizards. This version is a distinct improvement. I especially applaud the authors for adding the ANCOVA to their analyses. However, I still fail to see a strong connection between sympatric coexistence and potential thermal responses to climate change. If the authors feel that this connection is important, they should introduce the topic of sympatric coexistence in the Introduction and not wait until the Discussion to do so.

Response: Thanks for the reviewer’s comment. We've removed the relevant content and optimized the text. In addition we have changed the topic of sympatric coexistence to the introductory section. “Ecological niche segregation is a key mechanism for species coexistence and biodiversity. Sympatric species often differentiate through behavioral, morphological, or ecological traits, reducing competition by minimizing niche overlap [27]. With global climate change intensifying, understanding how sympatric species cope with temperature extremes has become increasingly critical [28]” (L111-115).

Comment2. L17: change “Due to the” to “Due to “their”

Response: Thanks for the reviewer’s comment. We have changed “Due to the” to “Due to their”. “Due to their thermosensitivity and ecological significance,…” (L17).

Comment3. L34-38: This sentence is long and confusing with too many comma-separated clauses.

Response: Thanks for the reviewer’s comment. We have modified the sentence. “In this study, the effects of temperature on the thermal biology and locomotor performance of two sympatric desert lizards, Eremias roborowskii and Phrynocephalus axillaris were examined. We analyzed morphological differences, the relationship between environmental temperatures (Te) and selected body temperatures (Tsel), and locomotor performance across varying Te. We also assessed critical thermal maximum (CTmax) and active body temperature (Tb) to evaluate current thermal conditions” (L31-37).

Comment4. L44: and other places. Never begin a sentence with an abbreviation.

Response: Thanks for the reviewer’s comment. We have revised and checked the full text (L44, 156, 319, 327, 400, 404).

Comment5. L47: delete comma after “size”

Response: Thanks for the reviewer’s comment. We have modified this sentence by removing the superfluous comma. “These differences may be attributable to variations in body size and ecological strategies,...” (L47).

Comment6. L80-83: This is another cumbersome sentence. Try this: “Their thermoregulatory mechanisms that consist of both behavioral and physiological responses enable them to adjust their behavioral and/or physiological limits to compensate for thermal fluctuations in the environment.”

Response: Thanks for the reviewer’s comment. We have simplified the sentences in the article. “Their thermoregulatory mechanisms that consist of both behavioral and physiological responses enable them to adjust their behavioral and/or physiological limits to compensate for thermal fluctuations in the environment [14-15]” (L80-83).

Comment7. L85: add “in” after “extremes”

Response: Thanks for the reviewer’s comment. We have changed “extremes” to “extreme”. “...to reduce direct exposure to extreme environmental conditions” (L85).

Comment8. L90: Because “endotherms” is used in line 91, replace “Poikilotherms” with “Ectotherms,” which is the opposite of endotherms. Poikilotherm is the opposite of homeotherm.

Response: Thanks for the reviewer’s comment. We have changed “Poikilotherms” to “Ectotherms”. “Ectotherms have a weaker physiological thermoregulatory capacity than endotherms,...” (L90).

Comment9. L92-93: change to “Some reptile species are thermal compliant animals, whose body temperature fluctuates with ambient temperature changes, giving them environmental flexibility. In contrast,…”

Response: Thanks for the reviewer’s comment. We have changed to change to “Some reptile species are thermal compliant animals, whose body temperature fluctuates with ambient temperature changes, giving them environmental flexibility. In contrast,…” (L92-93).

Comment10. L103: change “have been” to “are”

Response: Thanks for the reviewer’s comment. We have changed “have been” to “are”. “that are distributed in the desert area...” (L102).

Comment11. L107: change “morphological” to “morphology”

Response: Thanks for the reviewer’s comment. We have changed “morphological” to “morphology”. “locomotor phenotypes, morphology” (L107).

Comment12. L108: change “include” to “includes”

Response: Thanks for the reviewer’s comment. We have changed “include” to “includes”. “Research on P. axillaris includes habitat suitability evaluation and corridor modeling construction [24]” (L108).

Comment13. L115-118: This sentence is not clear. The introduction of a new topic, i.e., convergence evolution, clouds the issue.

Response: Thanks for the reviewer’s comment. We have deleted the words “convergence evolution”. “Meanwhile, comparing the similarities and differences in behavioral thermoregulation of two sympatric lizards, which will provide a vital supplement for the study of reptile ecological adaptability in extreme desert environments under climate warming” (L120-123).

Comment14. L127: add “and” after “substrates”

Response: Thanks for the reviewer’s comment. We have added “and” after “substrates”. “simulated natural habitats with soil and sand substrates, and provided with sufficient food...” (L131-132).

Comment15. L150: change “lizards” to “lizard”

Response: Thanks for the reviewer’s comment. We have changed “lizards” to “lizard”. “2.3 Measurement of lizard temperature” (L155).

Comment16. L182: change “represent” to “represents”

Response: Thanks for the reviewer’s comment. We have changed “represent” to “represents”. “Tb represents the internal temperature of an animal measured in nature during its active period” (L187).

Comment17. L191: change “equilibrating” to “were equilibrated”

Response: Thanks for the reviewer’s comment. We have changed “equilibrating” to “were equilibrated”. “...ensure body temperature were equilibrated with the chamber temperature to the corresponding test temperature.” (L196).

Comment18. L199: change to “methods involving gently brushing the tail”

Response: Thanks for the reviewer’s comment. We have changed to “methods involving gently brushing the tail”. “If the lizards paused, non-invasive methods involving gently brushing the tail base were applied to encourage them to resume running,...” (L204)

Comment19. L205: change “lizards” to “lizard”

Response: Thanks for the reviewer’s comment. We have changed “lizards” to “lizard”. “A complete locomotor test was defined as the lizard fully traversing the runway and turning around” (L210-211).

Comment20. L246: change “Meanwhlie” to “Meanwhile”

Response: Thanks for the reviewer’s comment. We have changed “Meanwhlie” to “Meanwhile”. “Meanwhile the first three principal components...” (L248).

Comment21. L309: change “mechanisms” to “mechanism”

Response: Thanks for the reviewer’s comment. We have amended the paragraph and changed “mechanisms” to “mechanism”. “Ecological niche segregation is a key mechanism for species coexistence and biodiversity” (L111).

Comment22. L314: change to “change, it is becoming…”

Response: Thanks for the reviewer’s comment. We have amended the paragraph. “With global climate change intensifying, understanding how sympatric species cope with temperature extremes has become increasingly critical [28]” (L113-115).

Comment23. L327-328: This is a clause, not a complete sentence. Change to: “…daytime temperatures, particularly engaging in…”

Response: Thanks for the reviewer’s comment. We have changed to“…daytime temperatures, particularly engaging in…” “Both species are diurnal, exhibiting daily activity rhythms compatible with high daytime temperatures, particularly engaging in basking and foraging activities in the pre-noon hours” (L321).

Comment24. L366: change “reflecting” to “reflects” [Verbal forms ending in -ing are present participles and cannot be used as the primary verb to indicate action.]

Response: We're very thankful for your detailed review. We have corrected "reflecting" to "reflects" as you suggested. Your input greatly improves our manuscript, and we appreciate it. “The correlation coefficient (linear slope k) reflects the relationship between body temperature and Te.” (L360).

Comment25. L380-381: change to “However, once the optimal temperature is surpassed, behavioral performances of lizards declines rapidly.”

Response: Thanks for the reviewer’s comment. We have modified the sentence. “However, once the optimal temperature is surpassed, behavioral performances of lizards declines rapidly [55]” (L374-375).

Comment26. L382: change to “constraints” to “constrains”

Response: Thanks for the reviewer’s comment. We have changed “constraints” to “constrains”. “Furthermore, when the lizards’ morphological structure constrains the maximization of all types of locomotor performance, a possible trade-off among these locomotor capabilities emerges [56]” (L376).

Comment27. L383: change to “…performance, a possible trade-off among these…”

Response: Thanks for the reviewer’s comment. We have changed to “…performance, a possible trade-off among these…”. “Furthermore, when the lizards’ morphological structure constrains the maximization of all types of locomotor performance, a possible trade-off among these locomotor capabilities emerges [56]” (L376-377).

Comment28. L391: change “populations” to “individuals”

Response: Thanks for the reviewer’s comment. We have changed “populations” to “individuals”. “whereas individuals with greater endurance generally exhibit lower sprint speeds” (L384).

Comment29. L398: delete “For the number of pauses,” Begin the sentence with “Although…”

Response: Thanks for the reviewer’s comment. We have modified the sentence. “Although differences between the two species were not significant,...” (L392).

Comment30. L408: change “suggested” to “suggest”

Response: Thanks for the reviewer’s comment. We have changed “suggested” to “suggest”. “These findings suggest that the two lizards exhibit distinct temperature adaptations,...” (L402).

Comment31. L409: add “be” after “may”

Response: Thanks for the reviewer’s comment. We have added “be” after “may”. “which may be associated with differences in their ecological niches and physiological traits” (L403).

Comment32. L418: add “these” before “two”

Response: Thanks for the reviewer’s comment. We have added “these” before “two”. “Global warming and rising ambient temperatures may significantly impact the locomotor performance of these two sympatric lizards” (L436).

This manuscript is a resubmission of an earlier submission. The following is a list of the peer review reports and author responses from that submission.

Round 1

Reviewer 1 Report

Comments and Suggestions for Authors

This manuscript reports on a study of thermal relationship and locomotory performance of two lizard species (Eremias roborowskii and Phrynocephalus axillaris) from China. In my opinion, the study suffers from some critical flaws regarding the experimental design and the conclusions drawn. Of central importance, measurement of Tpref is not appropriate. Preferred body temperature (and selected body temperature or target body temperature) relate to the body temperature that the individual ectotherm chooses. Thus, in a lab setting, these measures are generally measured in a thermal gradient where the individual has a wide range of environmental temperatures to choose from, and the measured body temperatures are considered to be the temperatures selected by the individual. In this study ‘Tpref’ was measured in environmental chambers where there was no gradient, precluding the lizards from attaining their preferred body temperature. The relationship measured between the chamber temperature and the body temperature reveal that in this situation, both species were forced to poikilothermic (no thermoregulatory options) and deviations from isometry are probably due to evaporative water loss at high environmental temperatures and excess tissue temperature at low environmental temperatures.

L65: The Bogert Effect should be included here.

L73: The term poikilotherm should be included here.

L75: Poikilotherms do not always have strong environmental adaptability.

L82: ‘Stably distributed’ is a strange term. Reword. How long is an ‘extended period’? Can you state a time period? If not, this is a meaningless statement.

L84: Do not start a sentence with an abbreviation.

L85: Change ‘taxa’ to ‘species’.

L111: Change ‘Determination’ to ‘Measurement’.

L112: Change ‘an electronic an accuracy’ to ‘an electronic balance with a precision’. Change ‘weight’ to ‘mass’.

L115: Change ‘accuracy’ to ‘precision’.

L126: This methodology needs to be made more explicit. If you were measuring Tpref, this must have been in a gradient, but you state that it is a constant temperature incubator. This is confusing because later in this paragraph you are talking about the lizard selecting temperatures. This paragraph needs to be rewritten to make this all much clearer. You need to include details on the gradient, the temperature range, how often the lizard’s temperature was measured. Having a look at the results now I see that you have not measured Tpref. Tpref is the temperature that the lizard chooses to be at – but you have made these measures in a thermal chamber without a gradient.

L130: How was the thermocouple secured to the lizard’s body?

L134: Change ‘determine’ to ‘measure’. This paragraph also needs to be rewritten. What were the temperatures in the incubator? How fast did lizard body temperature rise? All these factors have an impact on CTmax and CTMin. The description of the methods are not detailed enough to assess how robust they are.

L142: Change ‘determine’ to ‘measure’.

L146: This paragraph needs more detail. How were the lizards captured? How rapidly were you able to measure body temperature? How did you select which lizards to capture? If you caught some immediately after they emerged from burrows, the body temperature would not be the preferred body temperature. This needs much more detail.

L151: Change ‘Determination’ to ‘Measurement’.

L155: ‘adaptation’ is the incorrect word here. ‘Adaptation’ refers to evolutionary changes through generations. You mean ‘acclimation’, but a lizard will not really acclimate in 2 hours. I think that what you should say is that the 2 hours resulted in body temperature equilibrating with the chamber temperature.

L156: What is a ‘grease marker’?

Table 1; first line: Change 0.000 to <0.00.

L192: If SVL is the covariate in the ANCOVA (to correct for body size for a relative comparison), the ANCOVA will not test for differences in the SVL. Reword.

L219: Your design for estimating Tpref did not include a gradient.

Discussion: The discussion should start with a paragraph that synthesizes the findings of the study. The first paragraph of this manuscript does not even mention anything about the study. This needs to be rearranged.

L266: This paragraph contains information that is more suited to the introduction. You need to be covering the results of your study.

Comments on the Quality of English Language

Tpref needs to be properly defined and the English needs to be made more explicit in places.

Reviewer 2 Report

Comments and Suggestions for Authors

Dear authors,

Thank you for your effort on this study. Here you can track my suggestions and comments for your manuscript!

Best,

This manuscript addresses “the impact of climate warming” on the thermal biology and locomotor performance of two sympatric desert lizards, Eremias roborowskii and Phrynocephalus axillaris. The study presents valuable insights into how morphological traits and thermoregulation strategies mediate locomotor abilities in extreme environments. The study has ecological and evolutionary implications, particularly for species adaptation and conservation under climate change. While the manuscript is valuable, some areas require refinement to enhance clarity, methodological rigor, and presentation.

Positive points

a) The focus on two sympatric desert lizards provides insights into ecological niche differentiation and adaptation strategies.

b) The study includes slightly detailed morphological, thermal, and locomotor performance data supported by statistical analyses.

c) Releasing the lizards after experiments is a good case for other researchers as not to “preserve them” after the study is completed.

Revision-needed points 

Minor issues:

Abstract

i) It is kindly recommended to avoid from complex sentences for better readability. For example:  "Our results demonstrate that as a result of variations in body size and ecological strategies, the behavioral thermoregulatory mechanisms of two lizard species are diverse and complex.” Instead of that, it is suggested: “Our findings reveal that body size and ecological strategies drive diverse and complex thermoregulatory mechanisms in the two lizard species.”

ii)  It would be better to give some specific results (i.e Tpref and CTmax) in the abstract.

iii) Page 1 - line 32: Correct "dispalyed" to "displayed”.

iv) Page 1 - line 34: Fix "wiht" to "with"

Introduction

i) Page 2 - line 73: remove the dot after the word “strategies”.

ii) Page 2 - line 77-79: is there any reference for this statement?: “Additionally, differences in life habits also influence thermoregulatory strategies in lizards, with diurnal lizards more inclined to actively regulate their body temperature through basking, while nocturnal lizards show more thermal compliance, and the efficiency of their thermoregulation is strongly correlated with the ambient temperature.”

Discussion

i) Page 10 - line 353-354: Is there any reference or your reliable data for these extreme surface temperatures?

ii) Figure 5: 1st box: Global climate change: letter "e" is missing.

Major issues

Introduction

The introduction provides a broad overview, however it looks pretentious to “examine the effects of climate change”. I kindly recommend you to emphasize their “vulnerability to climate change”. Because your study is not a direct related study for “the effects of climate change.” Therefore, to re-think the title of the manuscript will also be better.

Mat & Met

What is the reason to solely compare male lizards? Why didn't you also compare the female lizards? Because even if these species are sympatric, there might be different strategies between sexes (Because their families are different). 

Results

In Mat Met section (line 146-150: 2.3. Measurement of lizard temparature), you mentioned that you have field measurements of active body temperatures. However, there is no field data of these two species: Tb (active body temperature) and Te (environmental temperature). How did you collect this data? Please write informative paragraph for this situation in the Results section.

Locomotor Performance: Clarify how repeatability was ensured (e.g., whether the two tests per individual were averaged or only the best performance was used).

Discussion:

a) The comparison of sympatric lizards is really valuable if it was really examined. Please look at these recent studies that how they benefited from field and/or lab data for this comparison. Because yours are really old data to compare. Moreover, comparison for observed thermal and locomotor data would be much valuable.

i) Åžahin, M. K., & Kuyucu, A. C. (2021). Thermal biology of two sympatric Lacertid lizards (Lacerta diplochondrodes and Parvilacerta parva) from Western Anatolia. Journal of Thermal Biology101, 103094. They investigated the effects of several environmental factor for regulating their thermal biology.

ii) Cabezas-Cartes, F., Kubisch, E. L., Duran, F., & Boretto, J. M. (2023). Comparative thermal sensitivity of locomotor performance and vulnerability to global warming of two sympatric Phymaturus lizards from cold environments of Patagonia (Argentina). Biological Journal of the Linnean Society140(2), 261-276. Their lab and field experiments are really good for comparing your results.

iii) Žagar, A., Gomes, V., & Sillero, N. (2023). Selected microhabitat and surface temperatures of two sympatric lizard species. Acta Oecologica118, 103887. They emphasised the importance of surface temperatures for the niche differentiation. 

b)  Include a concise discussion on whether these lizards possess the potential to adapt to increasing temperatures over evolutionary timescales or if they are at risk of local extirpation due to their limited thermal tolerance.

c) The discussion on speed vs endurance trade-offs is compelling; however, incorporating additional examples from other reptile studies would further strengthen the argument and provide a broader comparative perspective.

Reviewer 3 Report

Comments and Suggestions for Authors

The manuscript (MS) by Zheng et al. presents information on the thermal biology of two sympatric lizards living in the desert region of the Turpan Basin in China. I think their results will make a valuable contribution to the knowledge base of these two species as well as to the physiological ecology of lizards more generally. However, I have some problems with the MS as presented.

1.      The title is misleading. The authors did not test the effect of global climate change on these lizards but simply determined certain features of these species’ thermal biology. Certainly, they could speculate how thermal biology of these species might be impacted by climate change, but they presented no actual directly relating climate change to either lizard species. For example, are there data on increasing temperatures in the Turpan region as implied in line 379? If so, there would need to be thermal data on these lizards prior to this period of climate change for comparison.

2.      I do not see the point in the rather comprehensive morphometric analysis. Clearly, some of the traits measured (e.g., limb length, tail length) are relevant to running speed, but most of the traits seem to have little relevance to the point of the study, i.e., thermal biology.

3.      The MS lacks focus at times and goes into different areas. For example, the MS begins with an Introduction that discusses appropriately the need to understand the thermal biology of organisms and how they might be able to handle climate change. On the other hand, the Discussion begins with what reads like a completely different Introduction about niche segregation and coexistence.

4.      For the morphometric analysis, I recommend running a MANCOVA to get an overall view of the difference between the two species. Also, running multiple tests on the same data as implied in lines 178-179 requires downward adjustment (e.g., Bonferroni correction) of the alpha level used for significance testing because multiple test on the same data increase the probability of committing a Type I statistical error, i.e., a false positive test.

I recommend that the authors break this MS in half and write up two different papers, one focused on niche segregation with their morphometric analysis, which shows differences related to foraging behavior, etc. The other can then focus sharply on the thermal biology of these two lizards. Both papers would be of interest.

Relatively minor changes follow with line numbers:

34        “with” is misspelled

67        Be careful using the terms adapt and adaptation when referring to behavioral acclimatization. Although the lay public might understand the terms to be interchangeable, to biologists, adaptation has an evolutionary context.

71        “weaker” than what?

84        Spell out the genus at the beginning of a sentence.

97        change “environment” to “environments”

105      change to “with sufficient food including mealworms and…”

112      change to “an electronic scale to an accuracy”

272      insert “be” after “might”

293      No significant differences means no differences. Therefore, the consistently higher values for one species may well be simple sample error. On the other hand, a significant difference might exist in nature but did not show up because of small sample sizes.

319      The sentence beginning “However…” is not a complete sentence.

344      insert “be” after “may”

345      This sentence uses past tense, but the following use present tense. Be consistent.

352      change “lizard” to “lizards”

359      no need to spell out the genus the second time

446      title needs to be in lower case to be consistent with the other citations

483      “Naturalist” needs to be capitalized.

506      “Ethologist” needs to be capitalized.

545      capitalize “Journal of Land Use Science”

There’s also a problem with indention formatting on some of the references.
